# Exploring Novel Pyridine Carboxamide Derivatives as Urease Inhibitors: Synthesis, Molecular Docking, Kinetic Studies and ADME Profile

**DOI:** 10.3390/ph15101288

**Published:** 2022-10-19

**Authors:** Ayesha Naseer, Faisal Abdulrhman Osra, Asia Naz Awan, Aqeel Imran, Abdul Hameed, Syed Adnan Ali Shah, Jamshed Iqbal, Zainul Amiruddin Zakaria

**Affiliations:** 1Research Institute of Pharmaceutical Sciences, University of Karachi, Karachi 75270, Pakistan; 2Department of Pharmaceutical Chemistry, Faculty of Pharmacy and Pharmaceutical Sciences, University of Karachi, Karachi 75270, Pakistan; 3Department of Civil Engineering, Umm Al Qura University, Makkah 21961, Saudi Arabia; 4Center for Advanced Drug Research, COMSATS University Islamabad, Abbottabad Campus, Abbottabad 22060, Pakistan; 5Department of Pharmacy, COMSATS University Islamabad, Abbottabad Campus, Abbottabad 22060, Pakistan; 6Department of Chemistry, University of Sahiwal, Sahiwal 57000, Pakistan; 7Faculty of Pharmacy, Universiti Teknologi MARA Cawangan Selangor Kampus Puncak Alam, Bandar Puncak Alam 42300, Selangor, Malaysia; 8Atta-ur-Rahman Institute for Natural Products Discovery (AuRIns), Universiti Teknologi MARA Cawangan Selangor Kampus Puncak Alam, Bandar Puncak Alam 42300, Selangor, Malaysia; 9Borneo Research on Algesia Inflammation and Neurodegeneration (BRAIN) Group, Faculty of Medicine and Health Sciences, University Malaysia Sabah, Jalan UMS, Kota Kinabalu 88400, Sabah, Malaysia

**Keywords:** semicarbazide, thiosemicarbazide, semicarbazone, thiosemicarbazone, urease inhibition, molecular docking, kinetic study

## Abstract

The rapid development of resistance by ureolytic bacteria which are involved in various life-threatening conditions such as gastric and duodenal cancer has induced the need to develop a new line of therapy which has anti-urease activity. A series of pyridine carboxamide and carbothioamide derivatives which also have some novel structures were synthesized via condensation reaction and investigated against urease for their inhibitory action. Among the series, 5-chloropyridine-2 yl-methylene hydrazine carbothioamide (Rx-6) and pyridine 2-yl-methylene hydrazine carboxamide (Rx-7) IC50 = 1.07 ± 0.043 µM, 2.18 ± 0.058 µM both possessed significant activity. Furthermore, molecular docking and kinetic studies were performed for the most potent inhibitors to demonstrate the binding mode of the active pyridine carbothioamide with the enzyme urease and its mode of interaction. The ADME profile also showed that all the synthesized molecules present oral bioavailability and high GI absorption.

## 1. Introduction

The organic molecules possessing imine groups are called Schiff’s bases. Imine compounds have a wide range of applications such as catalysts [1], dyes [2], components in the polymer industry [3], and stabilizers [4]. Moreover, they have proven potent biological actions including anti-bacterial [5], anti-fungal [6], anti-inflammatory [7], anti-cancer [8], and anti-tumor activities [9] which make them a very important class of organic compounds in pharmaceutical and medicinal chemistry, as imine groups, thiosemicarbazone, and semicarbazone have no cytotoxicity on normal cells both in vivo and in vitro [10,11,12,13]. 

Urease aminohydrolase, precisely called urease (EC3.5.1.5), is a nickel-dependent metalloenzyme that is widely spread in plants bacteria, fungi, and higher plants [14,15,16,17,18]. The urease enzyme catalyzes the degradation reaction of urea into NH_3_ and CO_2_ which increases the pH of the specified organ [19,20,21,22]. The hyperactivity of the urease enzyme results in the accumulation of NH3 which increases the pH and causes favorable conditions for ureolytic bacteria, i.e., *H*. *pylori*, which is the main causing agent of gastric and duodenal ulcers and also leads to cancer [23,24,25,26,27,28]. Ureolytic bacteria are also responsible for various other pathogenic conditions such as infectious kidney stones, gastroduodenal inflammation, urinary tract infections, pyelonephritis, etc. [16,29,30,31,32,33,34]. 

The available line of treatment has been proven to be ineffective for ureolytic bacterial infections due to the rapid development of resistance by microorganisms [35]. The synthesis of small molecules for modulating the urease activity is a successful therapeutic strategy for overcoming the complication associated with it [34] and our research group has been involved for the last decade in the development of effective and potent inhibitors of urease enzyme [19,36]. For the continuation of this study, a pyridine carboxamide and carbothioamide were synthesized by the condensation reaction of pyridine carboxamide and carbothioamide. The produced semicarbazone and thiosemicarbazone have been selected for their urease inhibition action and promising results have been achieved in comparison with the standard thiourea (Figure 1). 

## 2. Results and Discussion

### 2.1. Synthesis 

The series of pyridine carbothioamide and carboxamide is synthesized as presented in Figure 1. The semicarbazone and thiosemicarbazone derivatives are synthesized by reacting the different substituted pyridine carboxaldehyde with semicarbazide HCl and thiosemicarbazide, respectively, in the presence of an aqueous solution of sodium acetate. The respective aldehyde and carbazide solutions are prepared in ethanol; carbazide solution is added dropwise into the aldehyde while the flask is stirred continuously at room temperature. The product is precipitated out within 30–60 min with a good-to-excellent yield i.e., 80–99.9%. The obtained product is separated through filtration, and purification is undertaken via recrystallization from a suitable solvent, i.e., ethanol. The characterization study is performed for thiosemicarbazone and semicarbazone via different spectroscopic techniques, including infrared, mass spectrometry, ^1^H, and ^13^C nuclear magnetic resonance spectroscopy. A pyridine carbothioamide is discussed here for an explanation of spectral studies; for example, in IR spectra at 3268 and 3255 cm^−1^, two weak peaks appear due to secondary amine (NH_2_), the NHCS group appears at 3181 cm^−1^ with weak intensity, and the medium peak appears at 1622 cm^−1^ (C=N). Mass spectrometry gives the molecular ion and base peak for determination of mass, with the relative abundance of the fragmentation pattern. We observed the molecular ion peak at *m*/*z* 181 with a relative abundance value of 20.7, and the most abundant fragment appears at *m*/*z* 180 with a relative abundance value of 99.9. The other fragments appear at *m*/*z* 120 (48), 92.0 (32), 64.9 (28); ^1^H NMR describes the presence of protons in the structure. The aromatic ring proton appears at δ 7.3 (t, 1H), 7.8 (t, 1H), 8.27 (d, 1H), 8.5 (d, 1H), whereas the CH-N appears at δ 8.08 (s, 1H), NH appears at 11.65 (s, 1H), and the terminal amine appears at δ 8.19 (s, 1H); ^13^C NMR describes the presence of carbon atoms in the structure (100 MHz): δ 120.68, 124.56, 136.99, 142.98, 149.75, 153.8, 178.8 ppm. 

### 2.2. The In-Vitro Urease Activity of Pyridine Carboxamide and Carbothioamide and Semicarbazone

The synthesized pyridine thiosemicarbazone and semicarbazone derivatives (Rx-1 to Rx-12) were checked for in vitro urease enzyme inhibition. All the tested molecules exhibited potent inhibitory enzymatic action in comparison with the standard inhibitor thiourea which exhibited the IC_50_ value of 18.93 ± 0.004 µM. The difference in the inhibition potential is exhibited due to the position and type of the substitution on the pyridine ring. Among all the derivatives, Rx-6 was found to have the most promising inhibition potential against urease, having an IC_50_ value of 1.07 ± 0.043 µM, while Rx-7 was the second most potent candidate with an IC_50_ value of 2.18 ± 0.058 µM as compared with the standard drug. The other synthesized derivatives also exhibited good inhibition potential which is summarized in Table 1. The presence of electron-withdrawing substitution (Br, OCH_3_, F) at the ortho position on the pyridine ring of carbothioamide makes evident the selective inhibition against urease with IC_50_ values of 3.13 ± 0.034 µM, 4.21 ± 0.022 µM, and 4.93 ± 0.012 µM, whereas for carboxamide these are 14.49 ± 0.067 µM, 5.52 ± 0.072 µM, and 5.96 ± 0.005 µM for Br, OCH_3_, F_3_, respectively. The electron-donating substitution (CH_3_) at the ortho position of the pyridine ring of carbothioamide has shown specific inhibition with an IC_50_ value of 6.41 ± 0.023 µM, and for carboxamide an IC_50_ value of 3.41 ± 0.011 µM. The electron-donating Cl group at the meta position of the pyridine ring of carbothioamide has shown the most potent inhibition with an IC_50_ value 1.07 ± 0.043 µM, and for carboxamide it shows selective inhibition with an IC_50_ value of 4.07 ± 0.003 µM.

### 2.3. Molecular Docking Studies

The in vitro inhibition of urease was evaluated via performing molecular docking studies. Initially, the crystal structure of urease (4Gy7) was downloaded from the Protein Data Bank (https://www.rcsb.org/) [37]. To compute the docking, MOE Software (version 2019, Chemical Computing Group, Montreal, QC, Canada) was utilized for the preparation of chemical structures by using its builder tool. The energy minimization was carried out by applying the force field of the MMFF94X with an RMS gradient of 0.001 Kcal/mol/A2. The minimized chemical structure of the ligand was uploaded, and the receptor was prepared in the docking software (LeadI, Germany). The minimization energy was estimated through docking and approximately 50 poses were selected to study the binding interactions of the inhibitor with the active site of the enzyme. The interactions of amino acid residues with inhibitors were visualized through Discovery Studio Visualizer (version, 4.0, BIOVIA, San Diego, CA, USA) [38]. Figure 2 shows the putative mode of binding at the active pocket of urease (PDB ID: 4GY7). The most potent inhibitor among the newly synthesized pyridine carboxaldehyde derivatives RX-1, 3, 6, and 7 was selected for the molecular docking studies. All the synthesized compounds showed strong binding affinity at the active pocket of urease through the formation of hydrogen bonds and hydrophobic interaction such as van der Waals forces.

Figure 3 shows the two-dimensional interaction of amino acid residues of the active pocket with different atoms of the inhibitor Rx-6, where the green line depicts hydrophobic interaction i.e., Lys709, Phe712, and tyr32 exhibited van der Waals interaction including π–π stacking between Lys709 and Phe712 with the pyridine part of the inhibitor. The nitrogen atom of the pyridine ring, C=N, N-H, and NH_2_ were found to be involved in the hydrogen bonding with Lys709, Tyr32, Ala80, and Glu742 residue of the active site, respectively. The amino acid residue Lys709 exhibited a strong H-bond via -NH interaction with the nitrogen atom of the pyridine ring, whereas -OH moiety of Tyr32 formed a hydrogen bond with the N-atom of azomethine linkage. The oxygen atom of the carbonyl group of Ala80 showed strong dual hydrogen bonding with the -NH_2_/NH group of thioamide linkage. Another prominent H-bond was formed with amino acid residue Glu742 of the active pocket. The Rx-1, 3, and 7 two-dimensional interaction figures are presented in Appendix A which shows strong π–π interaction and hydrogen bonding among amino acids residues and different atoms of the potent inhibitors.

### 2.4. Enzyme Kinetics Study

For the investigation of the type of inhibition of urease, an enzyme kinetic study was conducted with the most potent compound, Rx-6, among the identified inhibitors of urease. In the kinetic assay, varying concentrations of 0 µM, 1.25 µM, 2.50 µM, 5 µM, 10 µM, and 20 µM of urea as substrate and tested compound concentrations of 0.10 µM, 0.20 µM, 0.40 µM, 0.80 µM, and 1.6 µM were employed. By calculating the Lineweaver–Burk plot, the competitive mode of inhibition was shown by the compound Rx-6. Hence, it is evident from kinetic studies that the substrate and Rx-6 competed to bind at the active pocket; however, Rx-6 showed strong binding at the active pocket of urease, which was validated through a molecular docking study [39]. The Lineweaver–Burk plot is shown in Figure 4.

### 2.5. ADME Profile 

The Swiss ADME web server [40] calculates physiological parameters such as ADME (Adsorption, Distribution, Metabolism, Excretion) and drug-likeness of synthesized derivatives of pyridine carboxaldehyde. The SMILE format of all the synthesized compounds was uploaded to the web software to predict the physiological and pharmacokinetics parameters, and the results were tabulated manually (Table 2). The aqueous solubility of the synthesized molecule logarithm of solubility (LogS) was calculated on the base of the ESOL method [41] and all the molecules exhibited favorable results. The LogP (logarithm of water-octanol partition coefficient) was calculated to predict the lipophilicity character and it should not be greater than 5. All our synthesized molecules’ values lie within the range. Lipinski’s rule of 5 suggests that molecules should not have hydrogen bond acceptors (HBAs) and hydrogen bond donors (HBDs) of more than 10 and 5, respectively, and that all the compound’s structures must comply with the requirements. The impact of polar fragmentation over the surface of the structure is calculated by measuring the tPSA (topological polar surface area) and it should not be more than 140 Å2 as a higher surface area will limit the penetration of the BBB (blood-brain barrier) and result in lower membrane permeability [42]. All the synthesized molecules possessed good tPSA as they were within the desired range, i.e., 80.37–104.62. Veber’s rule was used to calculate the oral bioavailability and the results suggested that all the molecules have good oral bioavailability [43]. According to Veber’s rule, tPSA should be less than or equal to 140 Å, and the total hydrogen bond acceptor and the donor should not be greater than 12. All these rules are used to predict the ADME profile of the molecules, and whether they are accepted, intermediate, or rejected due to undesirable parameters. In our tabulated results it is concluded that all the compounds fulfil the complete requirements of having a good ADME profile and can act as good candidates for the oral treatments of the diseases associated with overactivity of the urease enzyme.

## 3. Materials and Methods

All the starting materials used in the synthesis, including semicarbazide HCl, thiosemicarbazide, pyridine carboxaldehyde, and sodium acetate, were purchased from Sigma Aldrich Co. The analytical grade reagent and solvent were purchased locally. The monitoring of the reaction was undertaken via thin-layer chromatography by applying silica gel to 60 aluminum backed plates at 0.063–0.200 mm as the stationary phase; a 70:30 ratio of n-hexane and ethyl acetate was used as the mobile phase. The travel of the solute was monitored under ultra-violet light at 254 nm. The single-spot and sharp melting point was the initial marker for the purified obtained product. The spectral analysis was carried out by conventional methods which include IR, ^1^H, and ^13^CNMR by typical procedures. Bruker Vector-22 spectrometers, a Bruker spectrometer 400MHz, and a Finnigan MAT-321A Germany were used for recording the spectra of IR, NMR, and mass, respectively. The melting point of the synthesized compounds was recorded on a stuart^TM^ melting point SMP3. 

### 3.1. Chemistry 

#### 3.1.1. Synthesis of Pyridine Carboxamide and Carbothioamide

The ethanolic solution of semicarbazide and thiosemicarbazide was added to a round bottom flask sequentially and stirred at room temperature, adding different substituted pyridine carboxaldehyde and an aqueous solution of sodium acetate in a 1:1:1 ratio. The reaction product was precipitated out within 30 to 60 min in the purified form with a good-to-excellent yield [44]. The characterization data can be found in the Appendix A.

#### 3.1.2. Characterization Data of Pyridine Carboxamide and Carbothioamide

##### Pyridine-2-yl-methylene thiosemicarbazide (Rx-1) [45] 

Yield: 96%, melting point = 212–216 °C, elemental analysis (calculated): C = 46.65, H = 4.47, N = 31.09, IR (KBr cm^−1^) 3268, 3255 (NH_2_), 3181 (NHCS), 1622 (C=N); EI-MS: *m*/*z* (rel. abundance %) 181 (M^+^, 20.7), 180 (99.9) 120 (48), 92.0 (32), 64.9 (28); ^1^H NMR (500 MHz, DMSO-*d*_6_): δ 7.3 (t, 1H), 7.8 (t, 1H), 8.08 (s, 1H), 8.19 (s, 1H), 8.27 (d, 1H), 8.5 (d, 1H), 11.65 (s, 1H); ^13^C NMR (125 MHz, DMSO-*d*_6_): δ 120.68, 124.56, 136.99, 142.98, 149.75, 153.8, 178.8. 

##### 6-Methylpyridine-2-yl methylene hydrazine-1-carbothioamide (Rx-2) [46]

Yield: 83%, melting point = 197–199 °C, elemental analysis (calculated): C = 49.46, H = 5.19, N = 28.84, IR (KBr cm^−1^) 3244 (NH_2_), 3116 (NHCS), 1614 (CH=N); EI-MS: *m*/*z* (rel. abundance %) 194 (M^+^, 16.9), 194.1 (99.9) 134.2 (57), 106.1 (38); ^1^H NMR (500 MHz, DMSO-*d*_6_): δ 2.46 (s, 3H), 7.22 (d, 1H), 7.70 (t, 1H), 8.03 (s, 1H), 8.08 (d, 1H), 8.34 (s, 1H), 11.6 (s, 1H). ^13^C NMR (125 MHz, DMSO-*d*_6_): δ_c_ 23.18, 119.51, 124.97, 139.59, 140.56, 151.28, 163.58, 178.95. 

##### 6-Bromopyridine-2-yl methylene hydrazine-1-carbothioamide (Rx-3)

Yield: 90%, melting point = 192–194 °C, elemental analysis (calculated): C = 32.45, H = 2.72, N = 21.62, IR (KBr cm^−1^) 3118, 3174 (NH_2_), 2952 (CH=N), 1529 (CH=N); EI-MS: *m*/*z* (rel. abundance %) 259 (M^+^, 4.4), 258.2 (99.9), 198.1 (62), 90 (58); ^1^H NMR (500 MHz, DMSO-*d*_6_): *δ* 7.6 (d, 1H), 7.7 (t, 1H), 7.98 (s, 1H), 8.27 (s, 1H), 8.33 (d, 1H), 11.75 (s, 1H); ^13^C NMR (125 MHz, DMSO-*d*_6_): *δ* 119.89, 128.52, 140.26, 140.86, 141.18, 155.18, 178.93.

##### 6-Methoxypyridine-2-yl methylene thiosemicarbazide (Rx-4) 

Yield: 87%, melting point = 224–226 °C, elemental analysis (calculated): C = 45.70, H = 4.79, N = 26.65, IR (KBr cm^−1^) 3260, 3371 (NH_2_), 3161 (NHCS), 1611 (C=N); EI-MS: *m*/*z* (rel. abundance %) 210 (M^+^, 71.6), 150.2 (78), 109.1 (30), 92.1 (50); ^1^H NMR (500 MHz, DMSO-*d*_6_): *δ* 3.04 (s, 3H), 6.79 (d, 1H), 7.71 (t, 1H), 7.87 (d, 1H), 7.96 (s, 1H), 8.13 (s, 1H), 11.65 (s, 1H); ^13^C NMR (125 MHz, DMSO-*d*_6_): *δ* 54.92, 111.51, 114.32, 141.34, 142.90, 150.98, 163.88, 178.65.

##### 6-Trifluoromethyl pyridine-2-yl methylene hydrazine-1-carbothioamide (Rx-5)

Yield: 82%, melting point = 219–221 °C, elemental analysis (calculated): C = 38.71, H = 2.84, N = 22.57, IR (KBr cm^−1^) 3151 (NHCS), 3028–2990 (NH_2_), 1595 (CH=N); EI-MS: *m*/*z* (rel. abundance %) 249 (M^+^, 25), 248.2 (99.9), 188.1 (40), 160.1 (29), 140.1(24), 102.0 (16); ^1^H NMR (500 MHz, DMSO-*d*_6_): *δ* 7.86 (d, 1H), 8.08–8.13 (Q,2H), 8.35 (s, 1H), 8.49 (s, 1H), 11.82 (s, 1H); ^13^C NMR (125 MHz, DMSO-*d*_6_): *δ* 121.18, 123.98, 139.27, 140.89, 154.68, 179.04. 

##### 5-Chloropyridine-2-yl methylene hydrazinecarbothioamide (Rx-6)

Yield 85%, melting point = 244–246 °C, elemental analysis (calculated): C = 39.16, H = 3.29, N = 26.10, IR (KBr cm^−1^); 3261, 3372 ((NH_2_), 3160 (NHCS), 1609 (CH=N) EI-MS: *m*/*z* (rel. abundance %) 215 (M^+^, 22.7), 214.2 (99.9), 154.1 (50), 126.1 (32), 99.0 (34); ^1^H NMR (500 MHz, DMSO-*d*_6_): *δ_H_* 7.96–7.98 (d, 1H), 8.06 (s, 1H), 8.26 (s, 1H), 8.33–8.35 (d, 1H), 8.38 (s, 1H), 11.68 (s, 1H),^13^C NMR (125 MHz, DMSO-*d*_6_): *δ* 121.90, 131.65, 136.93, 141.59, 148.25, 152.48, 178.91. 

##### 2-Pyridine-2-yl-methylene hydrazine carboxamide (Rx-7)

Yield: 87%, melting point = 191–193 °C, elemental analysis (calculated): C = 51.21, H = 4.91, N = 34.13, IR (KBr cm^−1^)1687 (C=O), 1585 (CONH), 923; EI-MS: *m*/*z* (rel. abundance %) 164.14 (M^+^, 10), 120.1 (99.9), 92.1 (36), 65.1 (20); ^1^H NMR (500 MHz, DMSO-*d*_6_): *δ*_H_ 6.64 (s, 1H), 7.31–7.33 (m, 1H), 7.78–7.81 (t, 1H), 7.88 (s, 1H), 8.14–8.16 (d, 1H), 8.52–8.53 (d, 1H), 10.51 (s, 1H); ^13^C NMR (125 MHz, DMSO-*d*_6_): *δ*_C_ 120.62, 124.28, 137.34, 140.41, 149.55, 153.67, 157.31.

##### 6-Methylpyridine-2-yl methylene hydrazinecarboxamide (Rx-8)

Yield: 83%, melting point= 218–219 °C, elemental analysis (calculated): C = 53.92, H = 5.66, N = 31.44, IR (KBr cm^−1^) 3164 (NH), 1696 (C=O), 1589 (CONH), 932; EI-MS: *m*/*z* (rel. abundance %) 178 (M^+^, 5), 134 (96), 106.0 (64), 79 (18); ^1^H NMR (500 MHz, DMSO-*d*_6_): *δ* 2.45 (s, 3H), 6.60 (s, 1H), 7.17–7.18 (d, 1H), 7.66–7.68 (t, 1H), 7.83 (s, 1H), 7.93–7.95 (d, 1H), 10.48 (s, 1H); ^13^C NMR (125 MHz, DMSO-*d*_6_): *δ* 24.13, 117.79, 123.57, 137.50, 140.53, 153.15, 157.27, 158.06. 

##### 6-Bromopyridine-2-yl methylene hydrazinecarboxamide (Rx-9)

Yield: 89%, melting point = 231–233 °C, elemental analysis (calculated): C = 34.59, H = 2.90, N = 23.05, IR (KBr cm^−1^) 1705 (C=O), 1566 (CONH), 921; EI-MS: *m*/*z* (rel. abundance %) 244 (M^+^, 10), 200 (70.4), 172.1 (24), 91.1 (36); ^1^H NMR (500 MHz, DMSO-*d*_6_): *δ* 6.70 (s, 1H), 7.56–7.57 (d, 1H), 7.74–7.75 (t, 1H), 7.76 (s, 1H), 8.19–8.20 (d, 1H), 10.63 (s, 1H) ^13^C NMR (125 MHz, DMSO-*d*_6_): *δ*_c_ 119.97, 125.35, 128.3, 128.80, 139.05, 154.88, 157.41.

##### 6-Methoxypyridine-2-yl methylene hydrazine carboxamide (Rx-10)

Yield: 97%, melting point = 193 °C, elemental analysis (calculated): C = 49.48, H = 5.19, N = 28.85, IR (KBr cm^−1^) 1684 (C=O), 1566 (CONH), 1022; EI-MS: *m*/*z* (rel. abundance %) 194 (M^+^, 10), 150.1 (99.9), 94.1 (16); ^1^H NMR (500 MHz, DMSO-*d*_6_): *δ* 3.85 (s, 1H), 6.60 (s, 1H), 6.75–6.76 (d, 1H), 7.77 (s, 1H), 7.69–7.74 (m, 2H), 10.50 (s, 1H); ^13^C NMR (125 MHz, DMSO-*d*_6_): *δ* 54.22, 110.66, 113.35, 139.91, 140.26, 151.64, 157.16, 163.74.

##### 6-Trifluoromethyl pyridine-2-yl methylene hydrazine carboxamide (Rx-11)

Yield: 96%, melting point = 233–235 °C, elemental analysis (calculated): C = 41.39, H = 3.04, N = 24.13,IR (KBr cm^−1^) 1715 (C=O), 1591 (CONH), 1132; EI-MS: *m*/*z* (rel. abundance %) 232.2 (M^+^, 8), 188.1 (99.9), 160.1 (40), 140.1 (24); ^1^H NMR (500 MHz, DMSO-*d*_6_): *δ* 6.74 (s, 1H), 7.81–7.82 (d, 1H), 7.88 (s, 1H), 8.06–8.09 (t, 1H), 8.48–8.49 (d, 1H), 10.71 (s, 1H); ^13^C NMR (125 MHz, DMSO-*d*_6_): *δ* 40.24, 123.43, 138.73, 139.22, 139.22, 146.41, 154.90, 156.94.

##### 5-Chloropyridine-2yl-methylene hydrazine carboxamide (Rx-12)

Yield: 86%, melting point = 221–223 °C, elemental analysis (calculated): C = 42.33, H = 3.55, N = 28.21, IR (KBr cm^−1^) 1707 (C=O), 1593 (CONH_2_), 1024, 922; EI-MS: *m*/*z* (rel. abundance %) 200 (M^+^, 4), 154.1 (97.7), 126 (26), 99.0 (18); ^1^H NMR (500 MHz, DMSO-*d*_6_): *δ* 6.68 (s, 1H), 7.86 (s, 1H), 7.92–7.95 (d, 1H), 8.21–8.22 (d, 1H), 8.56–8.57 (m, 1H), 10.56 (s, 1H); ^13^C NMR (125 MHz, DMSO-*d*_6_): *δ* 121.30, 130.94, 136.82, 138.86, 148.05, 152.9, 156.8.

#### 3.1.3. Urease Enzyme Inhibition Assay

The indophenol method was adopted for the estimation of urease inhibition activity for synthesized compounds with minor modification [47,48]. In the assay, the total volume was kept at 100 µL which was composed of 10 µL urease having a concentration of 5 U/mL, 40 µL of phosphate buffer (0.01mM K_2_HPO_4_+ EDTA + LiCl_2_); pH was adjusted to 8.15 using 40 µL of phenol reagent (1%, phenol, and sodium nitroprusside) and alkali reagent. This alkali reagent was composed of 0.5% of sodium hydroxide and 0.1% of active chloride (NaOCl). For the determination of urease activity, the test compound and enzyme were incubated for 30 min at 37 °C in the assay buffer; then phenol/alkali reagent was added. All reactions were performed in 96 well plates in a triplicate manner. Absorbance was measured at a wavelength of 625 nm using the instrument FLUOstar^®^ Omega (BMG LABTECH, Ortenberg, Germany). Initially, the percentage inhibition was estimated for the test compounds and compared with a standard inhibitor of urease (thioureas). Those synthesized compounds that exhibited an inhibition >50% were further diluted for the estimation of IC_50_ values. GraphPad (PRISM 8.0, San Diego, CA, USA) was utilized for the calculations of IC_50_ values. The synthesized compounds were dissolved in absolute ethanol for the percentage inhibition assay whereas dilutions were prepared in deionized water by keeping the final concentration at up to 1%. The standard inhibitor (thioureas) was used as a positive control in the assay whereas 1% ethanol was employed as a negative control during the assay.

## 4. Conclusions

The series of pyridine carboxamide and carbothioamide derivatives were successfully synthesized to check their urease inhibition potency. Amongst all the derivatives, Rx-6 and Rx-7 exhibited the most potent results with IC50 = 1.07 ± 0.043 µM, 2.18 ± 0.058 µM, respectively. Docking studies were performed to examine the binding interaction which showed that H-bonding, π–π, and van der Waals interaction are involved in the urease enzyme inhibition of the promising candidate. The mode of interaction was also checked by performing a kinetic study and the in silico ADME profile was checked by Swiss ADME software.

## Data Availability

Data is contained within the article and Appendix A.

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
