# Peer review of "Exploring Novel Pyridine Carboxamide Derivatives as Urease Inhibitors: Synthesis, Molecular Docking, Kinetic Studies and ADME Profile"

_pharmaceuticals, 2022, doi:10.3390/ph15101288_

Round 1
Reviewer 1 Report
This is an interesting work although is full of linguistic problems that makes reading difficult.
I would like to see in the manuscript one or two representative proton NMR spectra as I noticed that recent work by N. Georgiou (is ignored in the manuscript) deals with confermotional properties of theiosemicarbazones (Molecules 2022, 27,2537).
Do you see also isomers? It is very difficult to detect from the spectra of supplementary material.
Author Response
Thank you for your review, please see attached response.

Reviewer 2 Report
The manuscript entitled, " Exploring Novel Pyridine carboxamide Derivatives as Urease 2 Inhibitors: Synthesis, Molecular Docking, Kinetic studies and ADME profile" describes the synthesis of urea and thiourea derivatives of pyridine and evaluated for urease activity.
The synthesis of compounds is simple and single step. All the compounds have been characterized and evaluated for urease activity. The paper can be improved by incorporating following comments.
1. The C13 NMR data of all the compounds need to be added.
2. The CHN elemental analysis need to provided for each compound.
3. Conclusion need to be elaborated.
4. SAR of compounds need to be discussed.
5. Over all the paper need to checked for English Language.
Author Response
Response to Reviewer 2 Comments
Point 1: The C13 NMR data of all the compounds need to be added.
Response 1: The rest of the compound’s 13C NMR data has been added except the Rx-2 and Rx-9 due to shortage of the time. All the other structure elucidation’s techniques such as IR, Mass and 1H NMR has been successfully performed and their results are positively interconnected for the clamied structure.
Point 2: The CHN elemental analysis need to provided for each compound.
Response 2: Thank you for your valubale suggestion but as of now we have succesfully performed IR, Mass , 1HNMR, 13C NMRstudies for structal elucidation and their results are corelated with each other so elemental analysis is not further needed but We will keep this point in our mind and diffently will performed elemental analysis in our upcoming projects.
Point 3 Conclusion need to be elaborated..
Response 3: As per your valuable suggestion the conclusion of the research article is elaborated.
Point 4: SAR of compounds need to be discussed.
Response 4: Thank you for your suggestion, more details of the SAR has been added in the manuscript.
Point 5: Over all the paper need to checked for English Language..
Response 5: we regeret there were problem with english. The paper has been carefully reviewed to improve the grammer and readibility.
Reviewer 3 Report
1. The introduction s rather brief and blunt. More rational is needed as to why the authors consider the new scaffold has been proposed. I understand that the group has previous experience in the field, but it is nowhere near clear why they choose the alteration of their previously validated scaffold.
2. Table 1 , 2 , figure 4 – settings/ graphics must be improved. Editing required.
3. Line 110 and elsewhere in the text in-vitro/ in-vivo in Italics.
4. The synthesis is rudimentary. Also the variety of structures obtained is very limited. In order to optime activity a basic SAR should be developed. For this you should have used aldehydes that contain the same substituent but at different positions on the nucleus.
5. All synthetized compounds must be docked! This would allow for a better SAR profile and a comparison between in vitro and in silico data. According to figure 3, most of the predicted binding takes place via elements of the common scaffold. As such, the relevance of the p substituent does not seem to be obvious. A substituent that is better capable of interacting with Phe 712 or Lys 709 should be inserted.
6. Considering the discussion part, much more effort should be invested. SAR should be more detailed, comparison with previous paper, either from the same groups or from different ones, must be inserted. Feel free to use https://www.nature.com/articles/s41598-020-65107-9 .
7. Conclusions – must be improved.
8. References - many are outdated.
9. Consider having the manuscript proofread by a native speaker.
10. Add a visual abstract.
Author Response
Response to Reviewer 3 Comments
Point 1: The introductions rather brief and blunt. More rational is needed as to why the authors consider the new scaffold has been proposed. I understand that the group has previous experience in the field, but it is nowhere near clear why they choose the alteration of their previously validated scaffold.
Response 1: As per valuable suggestion of worty reviewer introduction has been modified and more rational data has been added.
Point 2: Table 1 , 2 , figure 4 – settings/ graphics must be improved. Editing required.
Response 2: As per suggestion of reviewer graphic and setting of the above mentioned table and figure have been improved.
Point 3: Line 110 and elsewhere in the text in-vitro/ in-vivo in Italics.
Response 3: Thanks for your keen observation. The correction has been made.
Point 4: The synthesis is rudimentary. Also the variety of structures obtained is very limited. In order to optime activity a basic SAR should be developed. For this you should have used aldehydes that contain the same substituent but at different positions on the nucleus.
Response 4: We are thankful for reviwer comment. Currently study is going on in our laboratory. So keeping mind the reviewer comment, the reaction with aldehyde with different substituent will be carried out in our upcoming studies.
Point 5: All synthetized compounds must be docked! This would allow for a better SAR profile and a comparison between in vitro and in silico data. According to figure 3, most of the predicted binding takes place via elements of the common scaffold. As such, the relevance of the p substituent does not seem to be obvious. A substituent that is better capable of interacting with Phe 712 or Lys 709 should be inserted.
Response 5: Thank you for the valuable suggestions. To address this point , further docking of synthesized derivtives were carried out. Other than RX-6, the potent inhibitors of urease RX-1, RX-3 and RX-7 were selected for docking studies. Their binding interactions with residues of active site were analysed. Two dimentional interactions of RX-1, RX-3 and RX-7 have shown in the figures and inserted in the supporting information of the manuscript. All synthesized compounds exhibited strong inhibition with the IC50 values less than 10 µM therefore only most potent inhibitors RX-6, Rx-1, RX-3, RX-7 were selected for the moelcular docking studies. In the figure 3, the interactions shown by the elements of the common scoffold with residues of the active pocket have justified the approximately similar trend of urease inhibition by all synthesized compounds. As per suggestion of worthy reviewer, we are highly motivated to carry out docking studies for all synthesized derivatives and we keep this point in mind and in our upcoming studies we will carry out docking studies for the all synthesized compounds.( Please see the supporting information)
Point 6: Considering the discussion part, much more effort should be invested. SAR should be more detailed, comparison with previous paper, either from the same groups or from different ones, must be inserted. Feel free to use https://www.nature.com/articles/s41598-020-65107-9.
Response 6: As per your valuable suggestion, The discussion part is being improved and above mentioned paper is cited in the manuscript.
Point 7: Conclusions – must be improved.
Response 7: As per your suggestion conclusions is improved.
Point 8: References - many are outdated
Response 8: new references are cited.
Point 9: Consider having the manuscript proofread by a native speaker.
Response 9: we regeret there were problem with english. The paper has been carefully reviewed to improve the grammer and readibility.
Point 10 : Add a visual abstract.
Response 10: Graphical abstract is added
Added

Round 2
Reviewer 2 Report
Following are the comments regarding the revised manuscript:
1. Some comments of the reviewers have been resolved. To improve the paper, the following comments need to be answered:
2. The paper needs to be edited by professional English editing services.
3. CNH or HRMS of all compounds need to be added to the text and spectra need to be attached in a supplementary file.
4. C13 spectra of all compounds need to be added in the supplementary file.
5. 1H NMR spectra of RX-6 is missing in the supplementary file.
The manuscript can be accepted after all the comments have been resolved.
Author Response
1.The paper needs to be edited by professional English editing services.
Answer: Sorry we regret for grammatical error. Dr Adnan who is currently working as Assoc. Prof. at Faculty of Pharmacy, Atta-ur-Rahman Institute for Natural Products Discovery (AuRIns), Universiti Teknologi MARA where mode of teaching is english has reviewed the paper for language editing.
- CNH or HRMS of all compounds need to be added to the text and spectra need to be attached in a supplementary file.
Answer: Elemental data of CNH is added in the manuscript
- C13 spectra of all compounds need to be added in the supplementary file.
Answer: All compound C13 spectra is being added in the manuscript
- 1H NMR spectra of RX-6 is missing in the supplementary file.
Answer: 1H NMR spectra is being added in the supplementary file.

Reviewer 3 Report
The improvements are welcomed. However, future corrections are needed, especially concerning English.
Author Response
The improvements are welcomed. However, future corrections are needed, especially concerning English.
Answer: Sorry we regret for grammatical error. Dr Adnan who is currently working as Assoc. Prof. at Faculty of Pharmacy, Atta-ur-Rahman Institute for Natural Products Discovery (AuRIns), Universiti Teknologi MARA where mode of teaching is english has reviewed the paper for language editing.

Round 3
Reviewer 2 Report
All the comments of the reviewer have been answered and the manuscript has been revised according to the comments and suggestions.
All the data have been provided in the supplementary file.